# A regulatory circuit of two lncRNAs and a master regulator directs cell fate in yeast

Fabien Moretto[1], N. Ezgi Wood[2], Gavin Kelly [1], Andreas Doncic[2,3] & Folkert J. van Werven[1]

Transcription of long noncoding RNAs (lncRNAs) regulates local gene expression in eukaryotes. Many examples of how a single lncRNA controls the expression of an adjacent or nearby protein-coding gene have been described. Here we examine the regulation of a locus consisting of two contiguous lncRNAs and the master regulator for entry into yeast meiosis, *IME1*. We find that the cluster of two lncRNAs together with several transcription factors form a regulatory circuit by which *IME1* controls its own promoter and thereby promotes its own expression. Inhibition or stimulation of this unusual feedback circuit affects timing and rate of *IME1* accumulation, and hence the ability for cells to enter meiosis. Our data demonstrate that orchestrated transcription through two contiguous lncRNAs promotes local gene expression and determines a critical cell fate decision.

[1] The Francis Crick Institute, 1 Midland Road, London NW1 1AT, UK. [2] Department of Cell Biology, UT Southwestern Medical Center, 6000 Harry Hines Boulevard, Dallas, TX 75390, USA. [3] Green Center for Systems Biology, UT Southwestern Medical Center, 6001 Forest Park Road, Dallas, TX 75390, USA. Correspondence and requests for materials should be addressed to F.W. (email: folkert.vanwerven@crick.ac.uk)

Expression of long noncoding RNAs (lncRNAs) influences local gene expression[1–7]. This process is conserved across species. A widespread mechanism through which lncRNAs modulate the expression of adjacent genes is by means of transcription-coupled chromatin changes[2,7–11]. Numerous examples of how a single lncRNA alters the expression of a nearby gene have been described[1,2,6,12–15]. Whether loci of two or more contiguous lncRNAs exist and how they regulate local gene expression remains unexplored.

Meiosis is central to gametogenesis during which a diploid cell gives rise to haploid gametes[16]. In *Saccharomyces cerevisiae* or budding yeast, the decision to enter meiosis is controlled by the master regulator transcription factor, *IME1*[17–19]. Transcription of *IME1* is tightly controlled by mating-type and nutrient signals[19]. In the presence of nitrogen and fermentable carbon sources, *IME1* is repressed via PKA and TORC signaling pathways[20]. During nutrient starvation, however, expression of *IME1* is induced in diploid cells and as a consequence cells enter meiosis.

Transcription of lncRNAs governs mating-type control of entry into meiosis in yeast[2]. In cells with a haploid mating type, transcription of the lncRNA *IRT1*, encompassing almost the complete *IME1* promoter, represses *IME1* expression via transcription-coupled chromatin changes[2]. In diploid cells, the transcriptional activator of *IRT1*, *RME1*, is repressed by the **a**1α2 repressor complex. This ensures that only cells expressing both mating types (*MAT*a and *MAT*α) can undergo meiosis. Interestingly, a large fraction of yeast natural isolates displays Rme1 expression in diploid cells suggesting that *IRT1* may also be active in this cell type[21–23]. Previous work suggested that a second lncRNA is expressed further upstream in the *IME1* promoter directly adjacent to *IRT1*[2,24]. The purpose of this transcript is not understood.

Here we examine how a cluster consisting of two lncRNAs and *IME1* controls entry into meiosis. We find that transcription of two contiguous lncRNAs facilitates a regulatory circuit through which *IME1* promotes its own expression and meiotic entry. Our results demonstrate how a locus of contiguous lncRNAs can interact in a non-intuitive manner to define a positive feedback loop that drives the decision to enter an important cell differentiation program. The work broadens the spectrum by which transcription of lncRNAs controls local gene expression.

## Results

**Two contiguous lncRNAs are expressed in the *IME1* promoter.** Previous work showed that in cells with a single mating-type *IME1* expression repressed by transcription through the *IME1* promoter of the lncRNA *IRT1*. A second lncRNA, named *MUT1523* or *IRT2*, annotated upstream in the *IME1* promoter directly adjacent to *IRT1*, and expressed in the same direction as *IME1* and *IRT1* was reported (Fig. 1a and Supplementary Fig. 1a)[2,24]. This transcript is about 400 bp and expressed in diploid cells during starvation. To examine whether *IRT2* is detectable by conventional northern or reverse transcription (RT)-PCR methods, we first measured its expression pattern in diploid cells of strain backgrounds that undergo meiosis proficiently (SK1) and poorly (S288C)[21]. We used SK1 because cells from this strain background enter meiosis synchronously, which makes the use of population-based assays possible for the study of meiotic regulatory mechanisms. In SK1, *IRT2* was detectable by northern blot in diploid cells exposed to sporulation medium (SPO), which induces cells to enter meiosis (Fig. 1b and Supplementary Fig. 1a–c). When we further examined the expression pattern in relation to the meiotic program, we found that *IRT2* was expressed prior and during meiotic divisions (Supplementary Fig. 1b, c). In S288C, *IRT2* expression was also clearly detected at

8 and 24 h in SPO by RT-PCR (Fig. 1c). We conclude that a second lncRNA, *IRT2*, is expressed in the *IME1* promoter in diploid cells during meiotic entry.

**IRT1 and IRT2 control meiotic entry.** The expression of *IME1* is critical for entry into meiosis in yeast[17,18]. Diploid cells that do not induce *IME1* expression will not enter meiosis. Given that *IRT1* and *IRT2* are localized in the *IME1* promoter, we next examined how their expression affects the propensity for cells to enter meiosis. We generated strains where *RME1*, the activator of *IRT1*, was deleted (*rme1Δ*) and where *IRT2* was expressed from an inducible promoter (*pCUP-IRT2*, +Cu)[2,25] (Fig. 1d). This set of mutants allowed us to dissect how *IRT1* and *IRT2* control entry into meiosis. In S288C, the fraction of cells that completed at least one meiotic division increased in the *rme1Δ* (35% compared to 21% for the control) suggesting that *IRT1* is also active in diploid cells (Fig. 1e). Furthermore, when *pCUP-IRT2* was expressed, a comparable increase in meiotic cells was observed (Fig. 1e). Conversely, in the absence of *IRT2*, (*pCUP-IRT2*, −Cu) the fraction of cells that underwent meiosis decreased (11% compared to 21% for the control). These data show that expression of *IRT1* represses meiosis, whereas expression of *IRT2* promotes meiosis.

In budding yeast, the a1α2 repressor complex, expressed in diploid cells from the opposite mating-type loci (*MAT*a and *MAT*α), represses *RME1* and accordingly *IRT1* expression with variable degree among yeast isolates[22,23]. SK1 harbors the *RME1* (del-308A) allele relative to S288C, which creates an additional canonical a1α2 repressor binding site[21]. This causes efficient *RME1* repression and thereby increases the ability of SK1 diploid cells to enter meiosis. Hence, *rme1Δ* had a minor effect on meiosis in SK1 (Fig. 1f). Induction of *pCUP-IRT2* expression also marginally affected meiosis in SK1 cells suggesting that *IRT1* is needed for *IRT2* function. To examine the relation between *IRT1* and *IRT2* further, we mutated the a1α2 repressor binding sites in the *RME1* promoter (*RME1-H*) in SK1 cells, which de-represses *RME1* expression[2,25] (Fig. 1d). Consequently, *IRT1* expression was increased and the fraction of cells entering meiosis reduced (75% compared to 14% for the control) (Fig. 1f and Supplementary Fig. 2a). Cells harboring *RME1-H* completed meiosis with a reduced efficiency when *IRT2* was repressed (*pCUP-IRT2*, −Cu) compared to cells harboring the wild-type *IRT2* promoter (4% compared to 14% for the control) (Fig. 1f). Importantly, induction of *IRT2* (*pCUP-IRT2*, +Cu) suppressed the meiotic phenotype caused by the *RME1-H* almost completely (70% compared to 14% for the control). It is worth noting that deleting *IRT2* also disrupted the *IRT1* promoter, and therefore was not suitable for studying *IRT2* function on its own (Supplementary Fig. 2b). We conclude that inhibitory effect of *IRT1* expression on meiosis can be suppressed by expressing *IRT2*.

**IRT2 promotes IME1 expression by repressing IRT1.** Having established that *IRT2* suppresses the effect of *IRT1* on meiotic entry, we next examined how *IRT2* influences *IRT1* and *IME1* expression directly. Given that the *IRT2* transcript covers two Rme1-binding sites directly upstream of *IRT1*, we hypothesized that *IRT2* transcription interferes with the binding of Rme1 and thereby represses *IRT1* (Fig. 1a). To test this directly, we expressed *IRT2* from an inducible promoter (*pCUP-IRT2*) in cells that also express Rme1 (*RME1-H*) (Fig. 1d). We found that in the absence of *IRT2* (*pCUP-IRT2*, −Cu), *IRT1* levels were constant, whereas transcription of *IRT2* strongly inhibited *IRT1* accumulation (Fig. 2a). Rme1 almost completely disassociated from the *IME1* promoter in the presence of *IRT2* (Fig. 2b). As a consequence, *IME1* expression was induced, cells underwent meiosis with faster kinetics, and more cells completed meiosis

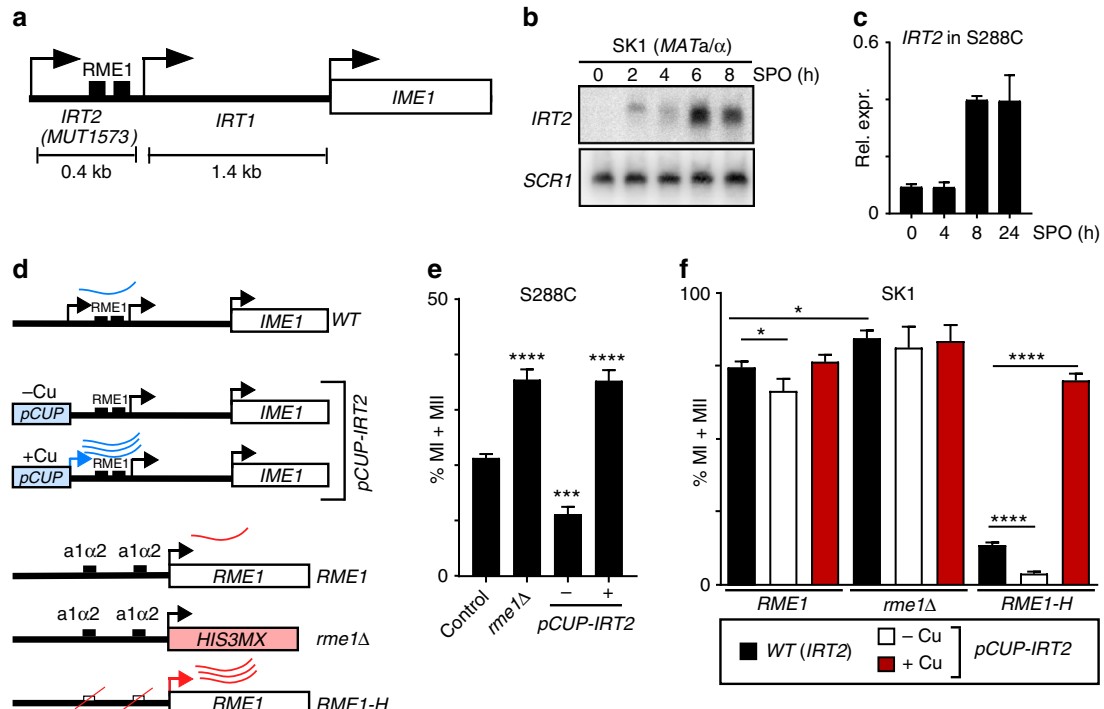

**Fig. 1** Transcription of *IRT2* promotes entry into meiosis. **a** Scheme of the *IME1* locus consisting of: *IRT1, IRT2/MUT1573*, and the *IME1* gene. **b** *IRT2* expression in SK1 diploid cells (FW1511) during entry into meiosis. Cells were grown in rich medium till saturation, shifted and grown in pre-sporulation medium for another 16 h, and transferred to sporulation medium (SPO). Samples for northern blot were taken at the indicated time points. A probe directed to the upstream region in the *IME1* promoter was used to detect *IRT2*. To control for loading, membranes were also probed for *SCR1*. **c** *IRT2* expression in S288C diploid cells (FW631) during entry into meiosis. Cells were grown till saturation in rich medium, and subsequently shifted to SPO. Samples were taken at the indicated time points. *IRT2* levels were quantified by reverse transcription and quantitative PCR. The signals were normalized to *ACT1* levels. The means ± SEM of $n = 2$ experiments are shown. **d** Scheme of *IME1* and *RME1* alleles used in **e** and **f**. **e** Quantification of cells that completed meiotic divisions (MI + MII) in diploid S288C cells that were either wild type (FW631), expressed *IRT2* from the inducible *CUP1* promoter (*pCUP-IRT2*) (FW2668) or harbored a deletion of *RME1* (*rme1Δ*, FW1497). Cells were grown as described in **c**, at 2 h in SPO, *pCUP-IRT2* cells were either not treated (−Cu) or treated with copper sulfate (+Cu). Cells were fixed after 72 h in SPO, stained, and DAPI masses of $n = 200$ cells were counted. Cells with two or more masses were considered to have completed at least one meiotic division. Means ± SEM of $n = 5$ experiments are shown. ***$p < 0.0005$; ****$p < 0.0001$ (Student's *t* test). **f** Quantification of cells that completed meiotic divisions (MI + MII) in SK1 diploid cells that were either wild type (FW1511), *pCUP-IRT2* (FW5254), *rme1Δ* (FW2340), harbored deletions in the a1α2 repressor sites of the *RME1* promoter (*RME1-H*, FW1196), harbored *pCUP-IRT2* and *rme1Δ* (FW2476), or *pCUP-IRT2* and *RME1-H* (FW2385). Cells grown and treated as described in **e**. Means ± SEM of at least $n = 4$ experiments are shown. *$p < 0.05$; ****$p < 0.0001$ (Student's *t* test)

(100% versus 65%) (Fig. 2a, c). We further examined whether *IRT2* controls *IME1* expression in *cis* or in *trans*. We generated a heterozygous *IME1* allele diploid strain with one copy harboring *pCUP-IRT2* together with *ime1Δ* and the other copy expressing the wild-type *IME1*. Since expression of *IRT2* (*pCUP-IRT2*, +Cu) did not improve the kinetics of meiosis compared to no expression *IRT2*, we conclude that *IRT2* controls *IME1* expression in *cis* (Supplementary Fig. 3a–d). Finally, we obtained evidence that *IRT2* represses the expression of *IRT1* directly and not via Ime1. Given that *IRT2* expression leads to increased expression of *IME1*, it is possible that Ime1 itself is responsible for repression of *IRT1*. To test this, we measured *IRT1* expression in cells harboring a 3′ end mutation in *IME1* (*ime1-t*), which impairs Ime1 function, in the presence of transcription of *IRT2* (*pCUP-IRT2*, +Cu). We found that *IRT2* efficiently repressed *IRT1* expression despite the presence of *ime1-t* (Supplementary Fig. 3e). In conclusion, *IRT2* transcription represses *IRT1*, and thereby promotes *IME1* expression and entry into meiosis.

**Ime1 feeds back to its own promoter and activates *IRT2*.** *IRT2* expression occurs in diploid cells only during entry into meiosis indicating that its transcription is tightly regulated. To identify

transcriptional regulators of *IRT2* expression, we scanned the *IRT2* promoter for sequence motifs. An Ume6 binding site (tgggtggcta) was identified about 90 bp upstream *IRT2*, which corresponds to −2314 bp of the *IME1* AUG (Fig. 3a). Indeed, Ume6 directly binds to the *IME1* promoter as measured by chromatin immunoprecipitation (ChIP) (Fig. 3b). Cells lacking Ume6 motif did not show binding Ume6. Further, the binding of Ume6 was independent of *IME1* expression because in the *ime1-t* mutant Ume6 binding was not affected (Supplementary Fig. 4a). The sequence motif is also present in related *Saccharomyces* species (Fig. 3a). Ume6 normally functions as a transcriptional repressor, however, in the presence of Ime1, Ume6 turns into a transcriptional activator[26,27]. Importantly, the expression of genes required for meiosis is controlled by Ume6[26,28]. To examine how Ume6 and Ime1 control *IRT2*, we measured *IRT2* expression in cells containing a deletion of the Ume6 motif (*pIME1-Δu6*), *ime1-t*, or both modifications. In *pIME1-Δu6* cells, baseline *IRT2* expression was detected (Fig. 3c, compare lanes 1–3 with 4–6). No *IRT2* expression was observed in *ime1-t* cells, while the expression of the double mutant was comparable to *pIME1-Δu6* (Fig. 3c, compare lanes 7–9 with 10–12 and 4–6). If *IRT2* expression depends on Ime1 expression levels then induction of *IME1* leads to increased *IRT2* transcription. To test this, we

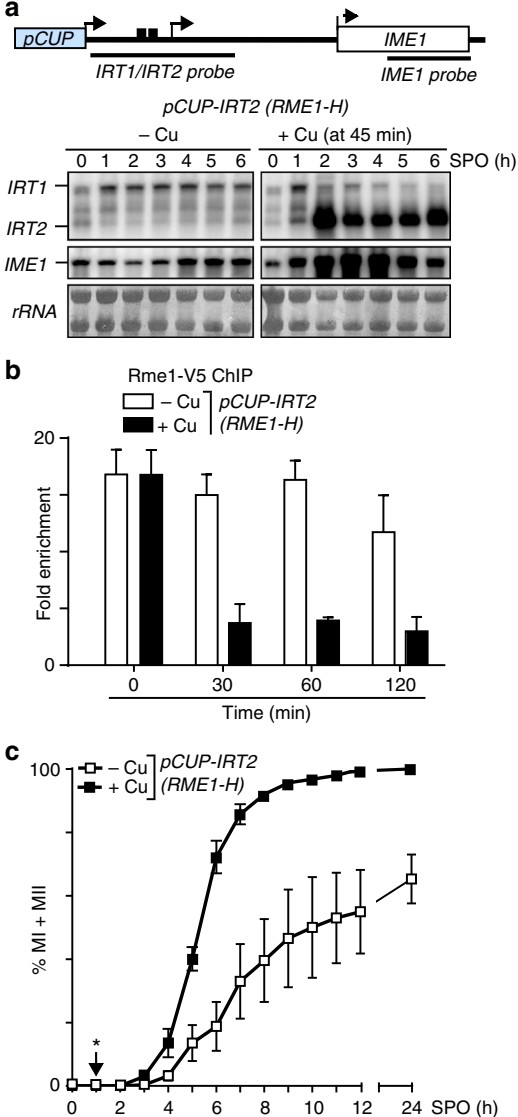

**Fig. 2** *IRT2* transcription interferes with Rme1 binding, and promotes *IME1* expression **a** *IRT2*, *IRT1*, and *IME1* expression in diploid cells harboring *pCUP-IRT2* and *RME1-H* tagged with the V5 epitope tag (FW2060) by northern blot. Cells were pre-grown in rich medium, pre-sporulation medium, before shifted to SPO and were either not treated (−Cu) or treated with copper sulfate (+Cu) after 45 min in SPO. Samples were taken at the indicated time points. Northern blot membranes were hybridized with a probe that detects both *IRT1* and *IRT2*, and a probe that detects *IME1*. As a loading control, the ribosomal RNA is displayed. **b** Similar as **a**, except that binding of Rme1 to the *IME1* promoter was determined by chromatin immunoprecipitation using V5-antibodies coupled to agarose beads in cells that were either not treated (−Cu), or treated with copper sulfate (+Cu). Samples were taken at the indicated time point after treatment, formaldehyde crosslinked, and chromatin extracts were prepared. Rme1-DNA complexes were isolated by immunoprecipitation, and the recovered DNA fragments were quantified by qPCR using a primer pair directed against the Rme1-binding sites in the *IME1* promoter. The signals were normalized to the silent mating locus (*HMR*), where Rme1 does not bind. The means ± SEM of $n = 4$ experiments are shown. **c** Similar as **a**, except that kinetics of meiotic divisions were determined. Means ± SEM of $n = 3$ experiments are shown. *Indicates the time of treatment with copper sulfate

measured *IRT2* expression in cells harboring *IME1* fused to an inducible promoter (*pCUP-IME1*). No detectable expression of *IRT2* was detected when *IME1* was not induced (Fig. 3d). Conversely, induction of *pCUP-IME1* (+Cu) was directly followed by increased *IRT2* expression, which required the Ume6-binding site (Fig. 3d and Supplementary Fig. 4b). In conclusion, Ume6 represses *IRT2*, whereas Ume6 together with Ime1 activate *IRT2* expression.

**Ime1 changes the local chromatin environment via *IRT2***. Having established that Ime1 promotes *IRT2* transcription, we next investigated how the feedback loop regulates the local chromatin environment. It is well-established that transcription-coupled chromatin changes repress cryptic and local gene expression[29]. To examine chromatin changes in context of Ime1-mediated *IRT2* transcription, we measured nucleosome positioning at the *IRT2* region using micrococcal nuclease (MNase) and quantitative PCR during entry into meiosis in absence or presence of Ime1 and *IRT2* expression. In the absence of Ime1 (*pCUP-IME1*, −Cu), the Rme1-binding sites were unprotected from the MNase digestion, which is indicative of an active promoter (Fig. 4a). In contrast, cells in which *IME1* was induced (*pCUP-IME1*, +Cu), and therefore *IRT2* was expressed, displayed stable nucleosomes around the Rme1-binding sites in the *IME1* promoter (Fig. 4a). This result indicates that transcription of *IRT2* establishes repressive chromatin to prevent Rme1 recruitment. Given that Rme1 promotes *IRT1* transcription and therefore an open chromatin state near the Rme1-binding sites, we next examined how Rme1 expression affects the nucleosome positioning. Interestingly, stable nucleosomes were also detected, but less prominent, in *RME1-H* cells indeed suggesting that Rme1 and *IRT2* have an opposing effect on nucleosome assembly in the upstream region of the *IME1* promoter (Supplementary Fig. 5a, right panel). In line with this observation, we found that Rme1 binding was strongly reduced when *IRT2* transcription was activated by Ime1 (Fig. 4b). As a consequence, *IRT1* levels were reduced (Fig. 4c, compare lanes 3–9 with 10–16).

Transcription-coupled chromatin changes are often accompanied with post-translational modifications of histones that have an important function in establishing repressive chromatin[29]. Previous work showed that in haploid cells *IRT1*-mediated repression of *IME1* requires Set2-dependent methylation of histone H3 lysine 36, and the histone deacetylase complex Set3C[2]. Haploid cells lacking Set2 and Set3 fail to repress *IME1* via *IRT1*, and undergo a lethal meiosis. In order to examine whether Set2 and Set3 also control *IME1* expression in diploid cells, we measured *IRT2*, *IRT1*, and *IME1* expression in *set2Δ set3Δ* cells harboring *pRME1-H*. As expected, *IME1* expression was increased in *set2Δ set3Δ* cells because *IRT1*-mediated repression of *IME1* is impaired in this mutant background (Fig. 4d). As a consequence, *IRT2* expression was also elevated (Fig. 4d, compare lanes 7–11 with 18–22). Conversely, *IRT1* expression decreased in the *set2Δ set3Δ* suggesting that *IRT2*-mediated repression of *IRT1* does not require Set2 and Set3 (Fig. 4d, compare lanes 7–11 with 18–22). In line with this observation, histone H3 lysine 36 methylation was not detected at *IRT2* while being transcribed (Supplementary Fig. 5b). *IRT2*-mediated repression of *IRT1* was also not impaired in *set2Δ set3Δ* mutant cells when we expressed *IRT2* from the *CUP* inducible promoter (*pCUP-IRT2*) (Supplementary Fig. 5c). We conclude that Set2 and Set3 regulate *IRT1*-mediated repression of *IME1* also in diploid cells, but are not important for *IRT2*-mediated repression of *IRT1*. Taken together, these results show that Ime1-mediated activation of *IRT2* changes the local chromatin environment.

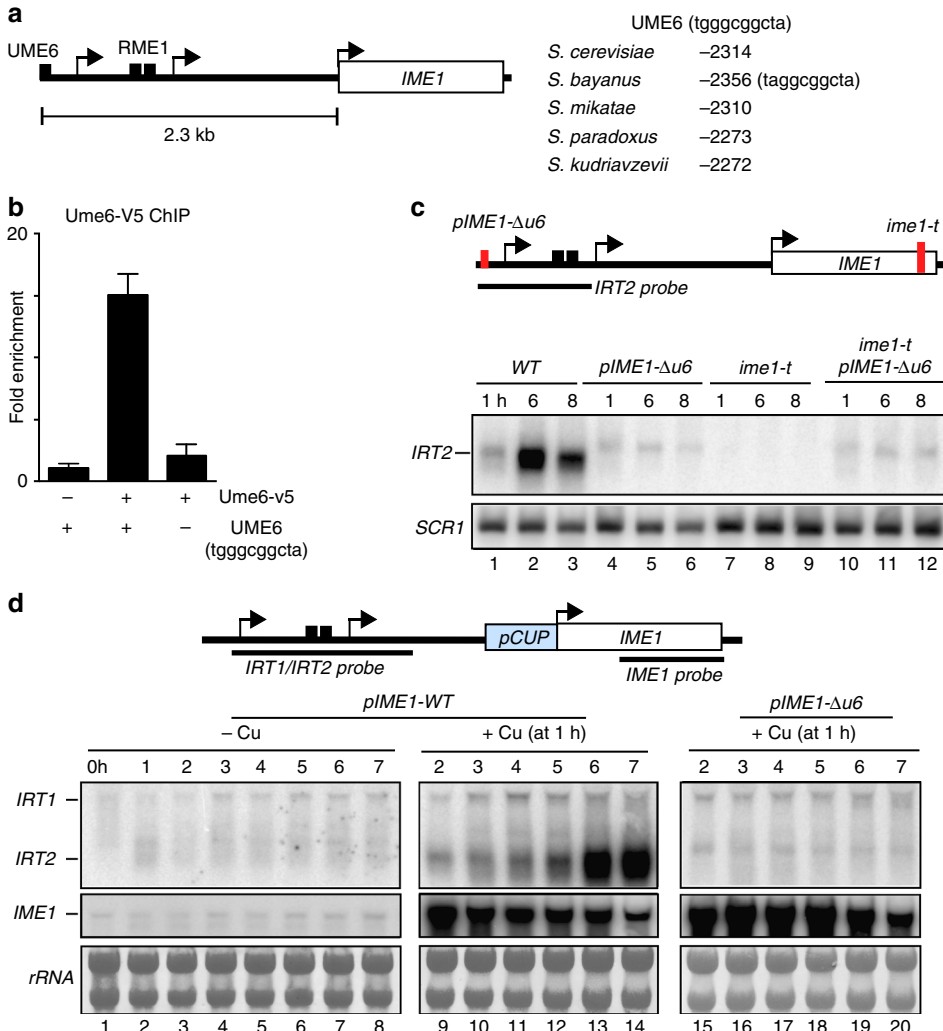

**Fig. 3** Ime1 and Ume6 control *IRT2* expression. **a** Schematic overview of the *IME1* locus indicating the position of the Ume6-binding site (left). Nucleotide coordinates of the Ume6 site relative to *IME1* AUG in different *Saccharomyces* species (right). **b** Binding of Ume6 to the *IME1* promoter measured by chromatin immunoprecipitation using V5 tagged Ume6 (FW2978). A wild-type (FW1509) and an Ume6-binding deletion mutant (*pIME1-Δu6*, FW2000) strains were also included in the analysis. The means ± SEM of *n* = 3 experiments are shown. **c** Northern blot of *IRT2* expression in cells that were pre-grown in rich medium and pre-sporulation medium, before shifted to SPO. Wild-type (FW1511), single, and double mutants harboring *pIME1-Δu6* and/or a 3′ end mutation in *IME1* (*ime1-t*) (FW2449, FW2370, and FW2571) strains were used for the analysis. To control for loading, the membrane was also probed for *SCR1*. **d** *IRT1*, *IRT2*, and *IME1* expression detected by northern blot in cells expressing *IME1* from the copper-inducible promoter (*pCUP-IME1*, FW3006). *pIME1-Δu6* (FW2842) cells were also included in the analysis. Cells were pre-grown in rich and pre-sporulation medium before shifted to SPO and were either not treated (−Cu) or treated with copper sulfate (+Cu) at 1 h in SPO. Northern blot membranes were hybridized with a probe that detects both *IRT1* and *IRT2*, and a probe that detects *IME1*. As a loading control, the ribosomal RNA is displayed

**Ime1 promotes its own expression**. Our observation that Ime1-mediated activation of *IRT2* represses *IRT1* expression predicts that Ime1 is able to promote its own transcription. To test this directly, we expressed one copy of *IME1* from an inducible promoter (*pCUP-IME1*), and inserted a superfolder green fluorescent protein (GFP) at the N-terminus (GFP-Ime1) of *IME1*, which allowed specific detection of *IME1* expression from the endogenous wild-type locus. The GFP-Ime1 signal increased during starvation showing that the construct is representative for *IME1* promoter activity (Supplementary Fig. 6a, b). When *pCUP-IME1* was induced (+Cu) in cells with *RME1-H*, GFP-Ime1 levels increased more than fivefold compared to non-induced control cells, which required the Ume6-binding site (Fig. 4e and Supplementary Fig. 6c). Deleting the Ume6-binding site (*pIME1-Δu6*) caused an aberrant expression of *IRT2* and an increased ability for cells to enter meiosis (Fig. 3c and Supplementary

Fig. 7a–c), which explained the increase in GFP-Ime1 levels in *pIME1-Δu6* cells compared to control cells (Fig. 4e). The results were comparable when we used cells harboring β-galactosidase *IME1* promoter fusion instead of GFP-Ime1 (Fig. 4f). Thus, Ime1 promotes its own expression in *trans*.

**Single-cell analyses of Ime1 feedback control**. Population-based assays could mask changes in expression within subpopulations of cells. To circumvent this possible caveat, we combined time-lapse microscopy with microfluidics and monitored *IME1* expression kinetics from a single locus (GFP-Ime1) in the absence or presence of the feedback control in single cells. The other *IME1* locus expressed mCherry from an *IME1* promoter lacking the *IRT2* sequence (*pIME1-Δirt2-mCherry*), which disrupts both *IRT1* and *IRT2* transcription. The *pIME1-Δirt2-mCherry* served

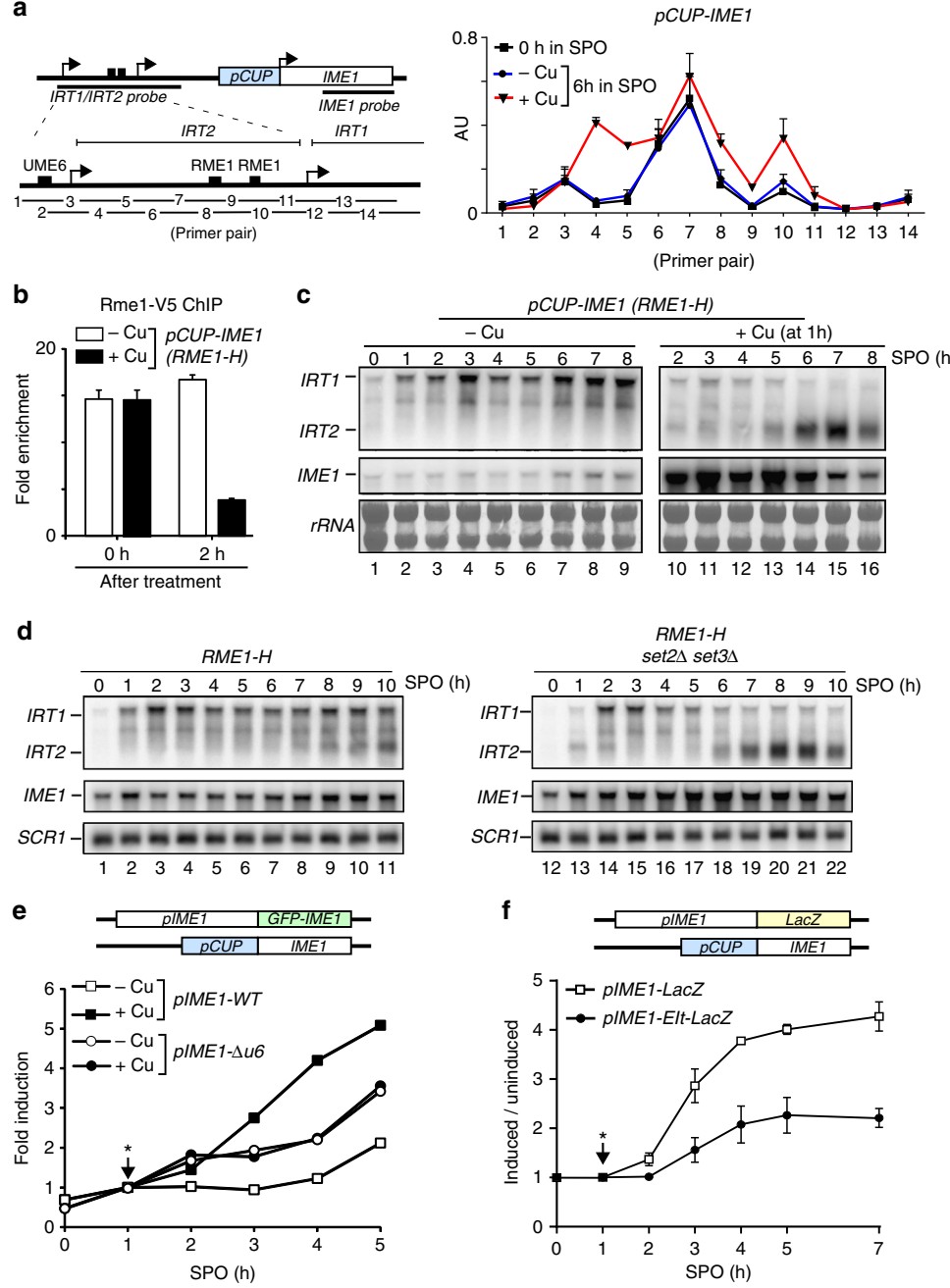

**Fig. 4** Ime1 changes local chromatin via *IRT2* and promotes its own expression. **a** Diploid cells harboring *pCUP-IME1* (FW3006) were grown in rich and pre-sporulation medium before shifted to SPO, and were either not treated (−Cu) or treated with copper sulfate (+Cu) after 1 h in SPO. Samples were taken at 0 and 6 h in SPO. Chromatin extracts were treated with micrococcal nuclease (MNase). Mononucleosome DNA fragments were isolated and quantified using 14 primers pairs. The signals were normalized to a no MNase input. Means ± SEM of *n* = 3. **b** Diploid cells with *pCUP1-IME1* and *RME1-H-V5* (FW1366) were induced to enter meiosis and were either not treated (−Cu) or treated (+Cu) after 1 h in SPO. Samples for chromatin immunoprecipitation were taken at 0 or 2 h after induction, and quantified by qPCR. The signals were normalized to the silent mating locus (*HMR*). Means ± SEM of *n* = 3. **c** *IRT1*, *IRT2*, and *IME1* expression in cells harboring *RME1-H* and *pCUP-IME1* (FW2270). Cells were grown as described in **a**. Northern blot membranes were hybridized with probes that detects *IRT1*, *IRT2*, and *IME1*. As a loading control, the ribosomal RNA is shown. **d** *IRT1*, *IRT2*, and *IME1* expression in diploid cells harboring *RME1-H* (FW1196) or *RME1-H* together with the *set2Δset3Δ* mutant (FW1312) during entry in meiosis. Cells were grown and samples were taken as described in **c**. Northern blot membrane was probed for *IRT1*, *IRT2*, and *IME1*. *SCR1* expression was used as loading control. **e** Cells containing one copy of *pCUP-IME1*, one copy of *pIME1-GFP-IME1* (*pIME1*) (FW5291), or *pIME1-Δu6-GFP-IME1* (*pIME1-Δu6*) (FW5295) were induced to enter meiosis in the absence (−Cu) or presence (+Cu) of *IME1* expression. Signals were quantified (*n* = 100 cells) per time point. Means ± error bars represent the 95% confidence interval. **f** Cells containing *RME1-H* and one copy of *pCUP-IME1*, one copy of *pIME1-LacZ* (FW5337), or a mutant *pIME1-Elt-LacZ* plasmid lacking part of the *IRT2* sequence including the Ume6 motif (FW5341) were induced to enter meiosis. The graph displays the ratio of β galactosidase activity signals from *IME1* induced samples versus not-induced samples. Means ± SEM of *n* = 3. *Treatment with copper sulfate

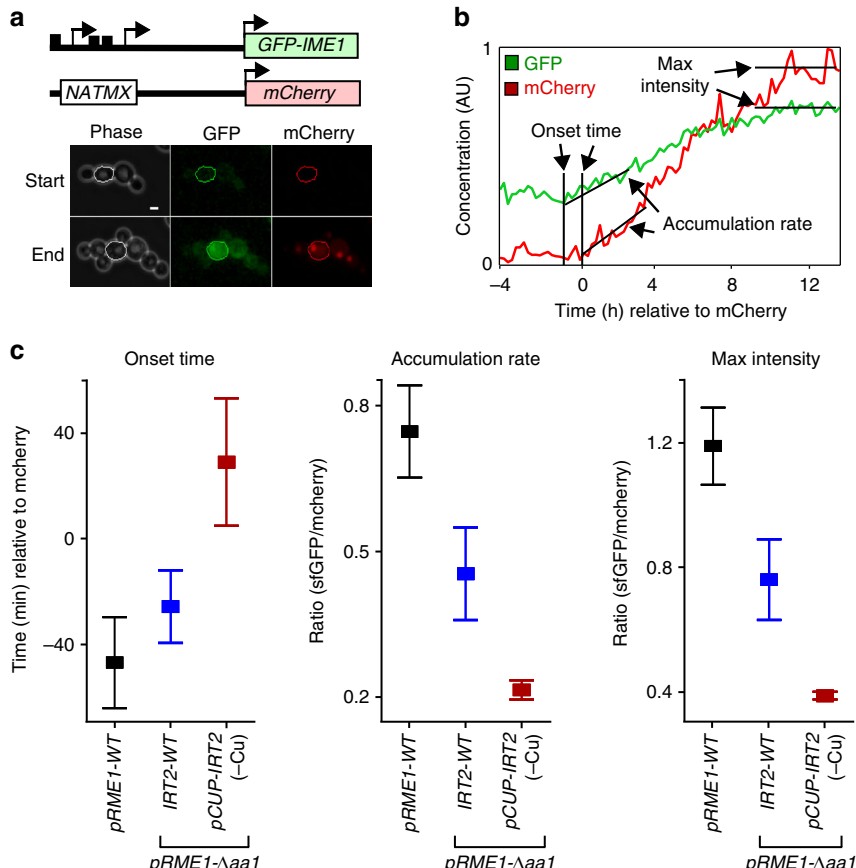

**Fig. 5** Single-cell analysis of the *IME1*, *IRT2*, and *IRT1* feedback cascade. **a** Schematic overview of *IME1* locus used for time-lapse microscopy (top). Diploid cells were heterozygous for the *IME1* locus: one copy of GFP fused to N-terminus of *IME1* (*pIME1-GFP-IME1*) and one copy expressing the *IME1* promoter fused to mCherry lacking the *IRT2* sequence (*pIME1-Δirt2-mCherry*). Cells were grown to log phase in synthetic complete media loaded into a microfluidic device, induced to enter meiosis and imaged for up till 50 h (see supplemental information for details). Example images of phase, GFP, and mCherry of a single cell taken at start and end of a time-lapse experiment (bottom). The scale bar in right bottom corner represents 3.2 μm. **b** Example of traces of a single cell. GFP and mCherry fluorescence signals were monitored and overall signals were scaled (see supplemental information for details). The graph also displays the time of initiation of expression (onset time), the rate of accumulation, and maximum levels (max intensity) of *pIME1-GFP-IME1* and *pIME1-Δirt2-mCherry*. **c** Summary of data obtained from time-lapse microscopy experiments. All strains harbored one copy of *pIME1-GFP-IME1* and one copy of *pIME1-Δirt2-mCherry*, combined with either wild-type *RME1* (FW4843), *pRME1-Δaa1* (FW4844), or *pRME1-Δaa1/pCUP-IRT2* (FW5051). Cells were imaged and GFP and mCherry fluorescence signals were quantified. Mean onset times, rate of accumulation, and max intensity were determined. The means ± SEM of n = 58 (FW4843), n = 88 (FW4844), and n = 64 (FW5051) cells are shown. All pairwise differences, except between FW4843 and FW4844 (left panel only), are statistically significant; *p* < 0.001 (Kolmogorov–Smirnov test)

as an internal control and was used for normalizing the GFP-Ime1 signal (Fig. 5a). In addition, we mimicked *RME1* expression of S288C by deleting only the proximal a1α2 repressor binding site in SK1 diploid cells (*pRME1-Δaa1*). Three parameters were measured from each single-cell GFP and mCherry trace: the time of *IME1* induction (onset time), rate of accumulation, and maximum expression of *IME1* (max intensity) (Fig. 5b and Supplementary Fig. 8). Cells containing *pRME1-Δaa1* had a lower *IME1* accumulation rate and lower max intensity compared to cells wild type for *pRME1* confirming that increased *IRT1* inhibits *IME1* (Fig. 5c). Importantly, cells where *IME1* was not able to feedback to its own promoter (*pCUP-IRT2*, −Cu) displayed a 60 min delay in onset of *IME1*, a twofold lower rate of accumulation, and a twofold lower max intensity. These data illustrate that the feedback loop consisting of Ime1, *IRT2*, and *IRT1* is critical for the timing of Ime1 induction and the rate of Ime1 accumulation.

**The Ime1 feedback loop is critical in yeasts expressing *RME1*.** Our data demonstrate that Ime1 promotes its own expression by activating *IRT2*, which in turn represses *IRT1* and thereby de-

represses *IME1*. Thus, in order for the feedback circuit to function, it requires expression of *IRT1* (Supplementary Fig. 2a). Indeed, the activator of *IRT1*, *RME1*, is expressed in a large proportion of diploid yeast isolates, including the S288C strain background[21–23]. To examine the physiological importance of the regulatory circuit consisting of Ime1, *IRT2*, and *IRT1* more closely, we measured *IME1* expression in S288C cells in the absence or presence of the feedback loop by modulating *IRT2* expression. Since S288C cells enter meiosis asynchronously, we monitored *IME1* expression using single-molecule RNA fluorescent in situ hybridization (smFISH) in single cells (Fig. 6a and Supplementary Fig. 9). When we repressed *IRT2* expression (*pCUP-IRT2*, −Cu), the fraction of cells expressing high levels of *IME1* (>40 copies per cell) decreased (2% compared to 13% for the wild type) (Fig. 6b). The fraction of cells harboring >40 *IME1* transcripts per cell increased when *IRT2* expression was elevated (*pIME1-Δu6*) (28% compared to 13% for the wild type) (Fig. 6b and Supplementary Fig. 7). The changes in the distribution of *IME1* expression among single cells were also reflected in the fraction of cells that completed meiosis (Fig. 6c). In the absence of *IRT2* expression (*pCUP-IRT2*, −Cu), the fraction of cells completing meiosis was

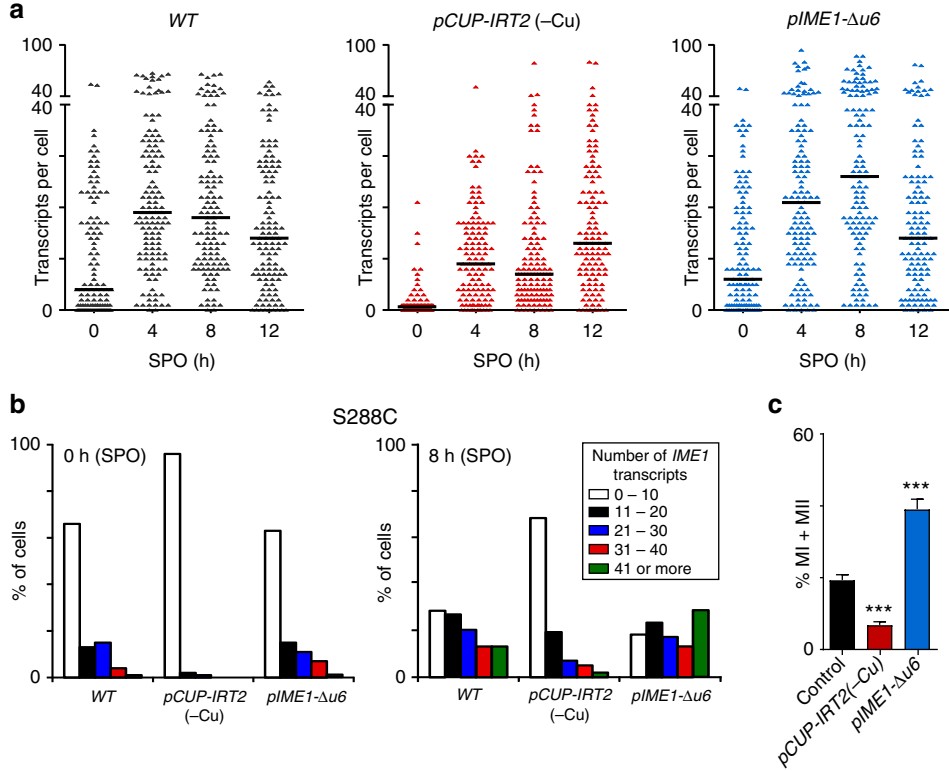

**Fig. 6** Distribution of *IME1* expression among single S288C cells in the presence or absence of *IRT2* transcription. **a** Distribution of *IME1* transcripts levels among single cells during entry into meiosis in S288C as detected by single-molecule RNA fluorescence in situ hybridization. Wild-type (WT) (FW631), *pCUP-IRT2* (FW2668), and *pIME1-Δu6* (FW1390) cells were grown in rich medium till saturation and subsequently shifted to SPO. Samples were taken at the indicated time points, fixed, and hybridized with probes directed against *IME1* and *ACT1* mRNAs (see supplemental information for details). The black line indicates the median number of transcripts per cell for each time point. At least *n* = 120 cells were used for the analysis. **b** Same as **a**, but cells were binned according *IME1* expression levels. **c** Quantification of cells that completed meiotic divisions in S288C cells described in **a**. Cells were fixed after 72 h in SPO, stained, and DAPI masses were counted. The means ± SEM of at least *n* = 5 experiments are shown. ***p < 0.0005 (Student's *t* test)

>50% reduced compared to wild-type cells (Fig. 6c). Conversely, a higher fraction of cells completed meiosis when expression of *IRT2* was increased (40% compared to 19% for the wild type). We conclude that the regulatory circuit consisting Ime1, *IRT2*, and *IRT1* directs the cell fate decision whether or not to enter meiosis.

**Model for *IME1* regulation by noncoding transcription**. Having established how transcription of *IRT2* and *IRT1* controls *IME1* expression and entry into meiosis, we next formulated our findings in a mathematical model with the aim to examine the dynamic properties of the regulatory circuit (Fig. 7, Supplementary Fig. 10 and Supplementary Note 1). Specifically, our model accounts for the dynamics of the transcription rates of *IRT1*, *IRT2*, and *IME1* as well as *IME1* mRNA and Ime1 protein levels (Fig. 7b and Supplementary Fig. 10a). The starvation signal, which is the primary signal for activation of *IRT1* and *IME1* transcription, was considered to be either off (0) or on (1) (Fig. 7b)[19,30]. We note that this simple model is sufficient to reflect the experimental data. In the absence of *IRT1* and *IRT2* transcription, Ime1 levels accumulated faster and were higher compared to the wild type, in line with the observations from the single-cell analyses (compare Fig. 5c (middle and right panel) to Fig. 7b (middle and right panel)). Conversely, in the absence of *IRT2* transcription, Ime1 levels were reduced compared to wild type. The model predicts a gradual increase in *IME1* transcription rates during starvation, which is in line with our observation that Ime1 can stimulate its own expression via *IRT2* and *IRT1* (Fig. 7b, middle panel).

Finally, we used our model to simulate how *IRT2* and *IRT1* control the dynamics of Ime1 accumulation during different periods of starvation (Fig. 7c and Supplementary Fig. 10b). We find that prolonged starvation (5 h) lead to almost maximum Ime1 expression for the wild-type *IME1* promoter, while a short period of starvation (1 h) resulted in low levels of Ime1 accumulation (Fig. 7c, compare left and right panels to Fig. 7b right panel). In the absence of *IRT1* and *IRT2*, a short period of starvation (1 h) displayed comparable Ime1 levels to a simulation of the wild type starved for 5 h (Fig. 7c, compare yellow left panel to blue line right panel). This suggests that the *IRT2-IRT1*-Ime1 regulatory circuit restricts Ime1 expression in cells exposed for starvation for short time periods. Taken together, we propose that there are possibly two functions for the feedback loop: (1) to elevate the rate of *IME1* transcription during prolonged starvation and (2) to prevent misexpression of Ime1 when the cells starved for a short period of time.

## Discussion

Cell fate decisions rely on precise expression of master regulatory genes. Here we described how in budding yeast a cluster of two lncRNAs is critical for controlling the expression of the master regulator for entry into meiosis, *IME1*. We demonstrate that orchestrated transcription of the two lncRNAs, *IRT2* and *IRT1*, define a feedback cascade that ensures timely induction of Ime1 and increases the rate of Ime1 accumulation (Fig. 7d). Our findings also illustrate how transcription of two lncRNAs transmits the signal of an upstream promoter element to the

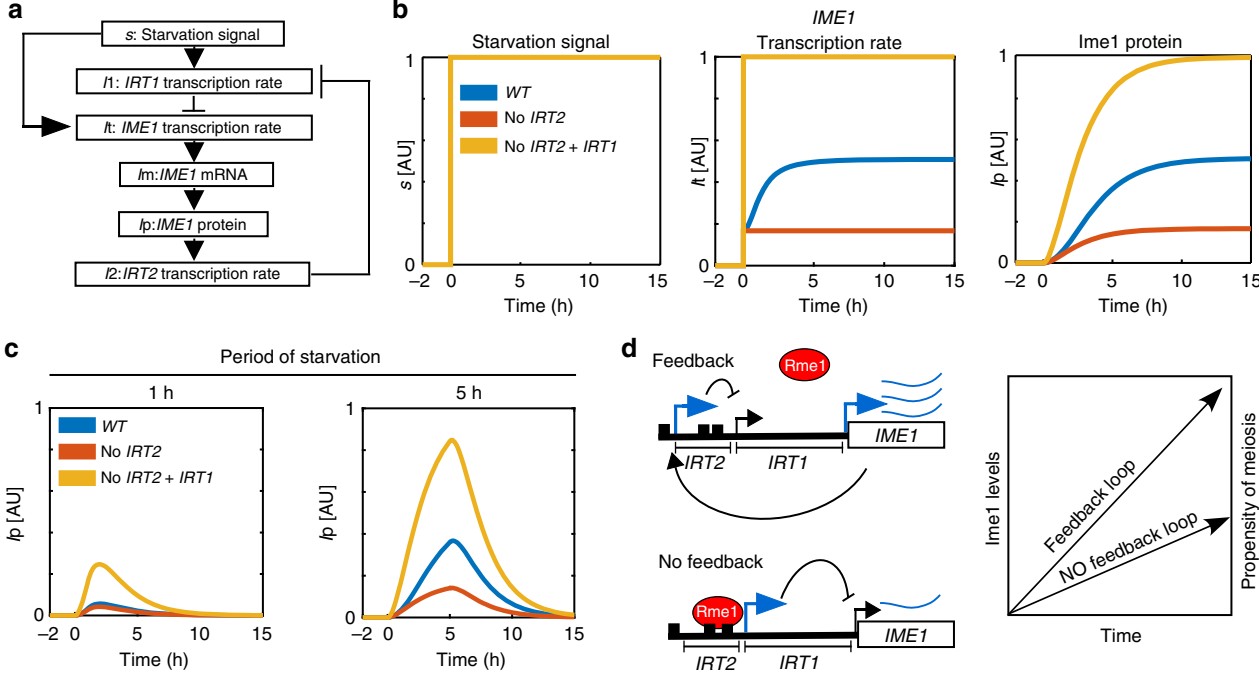

**Fig. 7** Model for the regulatory circuit consisting of *IME1* and the lncRNAs, *IRT2*, and *IRT1*. **a** Model for Ime1 regulation by *IRT1*, *IRT2*, and *IME1* (see Supplemental information for details). All the variables and connections between variables are displayed. **b** Simulation of *IME1* transcription rate (*I*t) and Ime1 protein accumulation (*I*p) prior (*s* = 0, before 0 h) and during starvation (*s* = 1) for the wild-type *IME1* promoter, in the absence of *IRT2* (no *IRT2*), or in the absence of *IRT1* and *IRT2* (no *IRT2* + *IRT1*). The y axis displays the level of *s*, *I*t, or *I*p in arbitrary units (AU) scaled between 0 and 1. **c** Similar as **b** except that Ime1 protein accumulation (*I*p) was simulated during starvation periods of 1 h and 5 h, respectively. **d** Model of feedback cascade involving *IME1*, *IRT2*, and *IRT1*. In the presence of the regulatory circuit consisting of *IME1*, *IRT2*, and *IRT1*, Ime1 is able to stimulate its own expression. This increases the propensity for cells to undergo meiosis. Cells lacking the feedback signals from Ime1 to *IRT2* and *IRT1* show reduced *IME1* expression and have a lower ability to enter meiosis

downstream gene. We propose that regulatory networks consisting of multiple contiguous lncRNAs can be used as a generic strategy for cells to control local gene expression dynamics.

Ime1 activates its own expression via transcription of the lncRNAs, *IRT1* and *IRT2*. Previous work showed that transcription of *IRT1* represses *IME1* in haploid cells by establishing repressive chromatin in the *IME1* promoter[2]. We found that *IRT1* is also active in diploid cells, yet the purpose *IRT1*-mediated repression of *IME1* in this cell type is not well understood. We propose that *IRT1* slows down meiotic entry till cells are "ready" to undergo meiosis. When cells are starved for nutrients, transcription of *IRT1* represses accumulation of *IME1*. As Ime1 levels increase, it is able to feed back to its own promoter. Together with Ume6, Ime1 activates transcription of *IRT2*, which in turn represses *IRT1* and consequently more Ime1 is produced. It has been shown that Ime1 and Ume6 can physically interact with each other[31,32]. The Ume6-Ime1 interaction is required for the activation of early meiotic genes[32]. Thus, only in the presence of Ime1 and Ume6, *IRT2* is transcribed and *IRT1* is repressed. In cells where Ime1 is not able to feed back to its own promoter, the expression of *IME1* is reduced and fewer cells enter the meiotic program. Thus, this regulatory switch ensures that Ime1 levels will increase only when cells are able to activate the expression of meiotic genes and progress into meiosis.

Why does Ime1 feed back to its own promoter via *IRT2*? Previous work showed that transcription of *IRT1* represses a relatively large region (over 1.5 kb) of the *IME1* promoter requiring the upstream Rme1-binding sites[2]. If Ime1 feeds back to its own promoter downstream of the Rme1-binding sites to directly activate its transcription, it would face interference by transcription of *IRT1*. Our data and modeling show that repression of *IRT1* transcription facilitates full activation of the *IME1*

promoter. We propose that transcription of *IRT2* is an effective way of repressing *IRT1*-mediated interference of the *IME1* promoter, and thereby promotes Ime1 accumulation and entry into meiosis.

A widespread mechanism for regulating gene expression locally is by transcription-coupled chromatin changes[2,3,6,9,10,15,29,33,34]. Previous work showed that transcription of *IRT1*, and not the *IRT1* RNA itself, is important for repression of the *IME1* promoter in haploid cells[2]. We propose that *IRT2* also uses a transcription-based mechanism for repressing *IRT1*. Like *IRT1*, transcription of *IRT2* alters the local chromatin structure (Fig. 4a), and inhibits binding of a transcriptional activator to the *IME1* promoter (Fig. 2b and Fig. 4b). In addition, we find that the repression of *IRT1* by *IRT2* works in *cis* only (Supplementary Fig. 3), suggesting that the *IRT2* RNA itself does not contribute to the repression mechanism. It is possible, however, that the *IRT1* and *IRT2* nascent transcripts contribute to repression of *IME1* and *IRT1*, respectively. For example, it was shown recently that histone methyltransferases, Set1 and Set2, interact with nascent mRNAs[35,36]. Interestingly, both enzymes are also important for *IRT1*-mediated repression of *IME1*, but not for repression of *IRT1* by *IRT2* transcription[2]. More work is needed to dissect roles of nascent RNAs and chromatin factors in facilitating the regulatory circuit consisting of Ime1, *IRT2*, and *IRT1*.

Our observations of the *IME1* locus raise the question how often transcription of multiple lncRNAs influences the expression of nearby coding genes. In budding yeast, cryptic unstable transcripts (*CUTs*) and stable unannotated transcripts (*SUTs*) are widely expressed[37,38]. Throughout meiosis, *CUTs* and *SUTs*, also known as meiotic unannotated transcripts (*MUTs*), are also transcribed extensively[24]. The *IRT1* and *IRT2* transcripts were identified as a *SUT* and a *MUT*, respectively. Given that the yeast

genome is gene-rich and *CUTs* are frequently transcribed in the opposite direction nearby of adjacent genes, it is not surprising that a high percentage of genes express at least one *CUT* or *SUT* within <400 bp distance of their coding sequence (Supplementary Fig. 11)[37,38]. Interestingly, we find that 8% of genes express more than two *CUTs* or *SUTs* within its proximity. Apart from *IME1*, the expression of at least two other genes is controlled by multiple lncRNAs, albeit by different mechanisms. At the *FLO11* locus, two overlapping lncRNAs are expressed adjacent to the *FLO11* gene, where they regulate a toggle switch for *FLO11* expression[39,40]. In fission yeast, a cascade of overlapping lncRNAs is required for the activation of fbp1 expression[5]. Taken together, regulation of local gene expression by multiple lncRNAs is not limited to *IME1* locus and may be widespread.

Like in yeast, in mammalian cells, lncRNAs are also expressed pervasively, and often nearby coding genes[41–43]. Several loci have been described where transcription of a single lncRNA regulates a nearby coding gene[15,44,45]. In addition, genome-wide approaches have also been used to identify loci where transcription of lncRNAs controls local gene expression[4,46]. Moreover, loci that express more than one lncRNAs also exits in mammalian cells. For example, the Myc oncogene expresses more than two lncRNAs upstream or downstream of its coding sequence[47–50]. While different functions have been attributed to the lncRNAs nearby the *Myc* gene, perhaps transcription of these lncRNAs influences Myc expression[47,48]. Thus, the basic features for regulating gene expression via transcription of multiple lncRNAs are present in mammalian cells. Dissecting the mechanisms of how local gene expression is controlled through transcription of lncRNAs in yeast may facilitate the understanding in higher eukaryotes.

## Methods

**Yeast strains and plasmids**. Yeast strains used in this paper were derived from the SK1 and the S288C strain backgrounds. The genotypes are listed in Supplementary Table 1. Gene or promoter deletions were generated using the one-step deletion protocol as described previously[51]. Strains harboring deletions in two or one a1α2-binding sites present in the *RME1* promoter resulting in the *RME1-H* and *pRME1-Δaa1* alleles, respectively, were described previously[2]. The *ime1-t* mutant harbors a truncation and a 3′ end mutation in *IME1* (L325A), which was obtained through one-step integration of a *NATmx* cassette resulting in a deletion from 50 bp of the 3′-end of *IME1* and two nucleotide changes within *IME1*. The strains harboring *RME1* with three copies of V5 at C-terminus was described previously[2]. Plasmid p158 harboring *pIME1-LacZ* was previously described[52]. Plasmid p260 lacking part of the *IRT2* sequence was generated by digestion of p158 with EcoRI and EcoNI, the ends were blunted and ligated together to generate *pIME1-EIt-LacZ*. The *IME1* promoter fused to *LacZ* strains in Fig. 4e and Supplementary Figure 6 were generated by linearizing the plasmids, p158 and p260, with StuI and integrated at the *URA3* locus. The *IME1* with the N-terminal superfolder GFP (GFP) was generated using a seamless tagging approach as described previously[53].

**Growth and conditions**. All experiments were performed at 30 °C in a shaker incubator at 300 r.p.m. Starvation-induced synchronous sporulation were previously described[54]. SK1 strains were grown till saturation for 24 h in YPD (1.0% (w/v) yeast extract, 2.0% (w/v) peptone, 2.0% (w/v) glucose, and supplemented with uracil (2.4 mg/l) and adenine (1.2 mg/l)), cells were then diluted at $OD_{600} = 0.4$ to pre-sporulation medium (BYTA) (1.0% (w/v) yeast extract, 2.0% (w/v) bacto tryptone, 1.0% (w/v) potassium acetate, 50 mM potassium phthalate) grown for about 16 h, subsequently centrifuged, washed with sterile miliQ water, centrifuged again, and re-suspended at $OD_{600} = 1.8$ in SPO (0.3% (w/v) potassium acetate and 0.02% (w/v) raffinose)). S288C strains were directly shifted from saturated YPD to SPO medium following the same procedure. For Fig. 1f and Supplementary Figure 7b, SK1 strains were induced to sporulate by shifting them directly from YPD to SPO following the same procedure.

Cells harboring *CUP1* promoter fused to *IRT2* or *IME1* (*pCUP-IRT2* or *pCUP-IME1*) were treated after 1 h in SPO with copper sulfate when cells were shifted from BYTA to SPO or treated after 2 h when cells were shifted from YPD to SPO. In Fig. 4e and Supplementary Figure 6, 5 μM of copper sulfate was used, whereas for other experiments using the SK1 strain background 25 μM was used. To induce *pCUP-IRT2* in the S288C strain background, 50 μM was used.

For the LacZ *IME1* promoter induction assay presented in Fig. 4f, cells were grown to saturation in YPD and diluted at $OD_{600} = 0.05$ overnight, then at around

$OD_{600} = 6–8$ cells were washed once with sterile millQ water and re-suspended in SPO at $OD_{600} = 1.8$.

For the live-cell imaging presented in Fig. 5 and Supplementary Figure 8, cells were grown to log phase in synthetic complete media (SCD) and loaded into a microfluidic device (Cellasic, Y04C plate) where they were re-suspended in SCD for an additional 2 h, after which we exposed the cells to YNA (2% potassium acetate, 0.25% yeast extract) for 50 h at constant flow rate (0.6 psi) at 25 °C temperature[55].

**Nuclei/DAPI counting**. DAPI staining was used to monitor meiotic divisions throughout time courses. Cells were fixed in 80% (v/v) ethanol, pelleted by centrifugation, and re-suspended in 100 mM phosphate buffer (pH 7) with 1 μg/ml 4′,6-diamidino-2-phenylindole (DAPI). Cells were then sonicated a few seconds and left in the dark at room temperature for at least 5 min. The proportion of cells containing one, two (meiosis I), three or four (meiosis II) DAPI masses were counted using a fluorescent microscope.

**RT-qPCR**. To quantify *IRT1*, *IRT2*, and *IME1* RNA levels in S288C and SK1 strain backgrounds, we used a reverse transcription combined with quantitative PCR (RT-qPCR). Total RNAs were isolated by acid phenol–chloroform extraction, ethanol precipitated, treated with DNAse, and further column purified (Macherey-Nagel). About 1 μg of total RNAs were reverse-transcribed using random primers and Protoscript II (NEB), and single-stranded complementary DNAs were quantified by real-time PCR using Express SYBR green mix (Life Technologies) on a 7500 Fast Real-Time PCR system (Life Technologies). The signals were normalized to *ACT1* mRNA levels. Oligo nucleotide sequences used for RT-qPCR experiments are displayed in Supplementary Table 2.

**Chromatin immunoprecipitation**. Chromatin immunoprecipitation experiments were performed as described previously[2]. Cells were fixed in 1.0% w/v formaldehyde for 25 minutes at room temperature and quenched with 100 mM glycine. Cells were lysed in FA lysis buffer (50 mM HEPES–KOH, pH 7.5, 150 mM NaCl, 1 mM EDTA, 1% Triton X-100, 0.1% Na-deoxycholate, 0.1% SDS, and protease cocktail inhibitor used as recommended by the manufacturer (complete mini EDTA-free, Roche)) using beadbeater (BioSpec) and chromatin was sheared by sonication using a Bioruptor (Diagenode, 8 cycles of 30 s on/off). Extracts were incubated for 2 h with 15 μl of anti-V5 agarose beads (Sigma) or 20 μl of magnetic Prot A beads (Sigma) coupled with 0.5 μg of a polyclonal antibody raised against Histone H3 (Ab1791, Abcam) or histone H3 trimethyl lysine 36 (Ab9050, Abcam), washed twice with FA lysis buffer, twice with wash buffer 1 (FA lysis buffer containing 0.5 M NaCl), and twice with wash buffer 2 (10 mM Tris-HCl, pH 8.0, 0.25 M LiCl, 1 mM EDTA, 0.5% NP-40, 0.5% Na-deoxycholate). Subsequently, reverse cross-linking was done in 1% SDS-TE buffer (100 mM Tris pH 8.0, 10 mM EDTA, 1.0% v/v SDS) at 65 °C overnight. After 2 h of proteinase K treatment, samples were purified and DNA fragments were quantified by real-time PCR using SYBR green mix (Life Technologies) using primers described in Supplementary Table 2. Signals were normalized over the *HMR* locus, which showed no binding for Ume6 or Rme1. For Supplementary figure 5b, H3K36me3 ChIP signals were normalized to histone H3 ChIP.

**Micrococcal nuclease digestion qPCR**. To determine the chromatin structure at the *IRT2* locus, mononucleosomes were extracted and purified using a MNase digestion protocol that was described previously[56]. Approximately, 160 $OD_{600}$ units of cells were crosslinked for 25 min at 30 °C and 250 r.p.m. with formaldehyde (1% v/v). Reaction was quenched with glycine (125 mM). Subsequently, cells were re-suspended in 20 ml of buffer Z (1 M sorbitol, 50 mM Tris-HCl pH 7.4) plus β-mercaptoethanol (10 mM) and treated with 250 μg of T100 Zymolase for 60 min. Next, cells were re-suspended in 1 ml NP buffer (0.5 mM spermidine, 1 mM β-mercaptoethanol, 0.075% (w/v) tergitol solution-type NP-40 detergent, 50 mM NaCl, 10 mM Tris-HCl pH 7.4, 5 mM MgCl₂, 1 mM CaCl₂), vortexed for 10 s, and 100 μl of extract was treated with 0.2 μl of MNase (2 mg/ml, NEB) for 30 min at 37 °C, the reaction was quenched with EDTA (10 mM), and reverse crosslinked overnight in 1% SDS-TE (100 mM Tris pH 8.0, 10 mM EDTA, 1.0% v/v SDS). Samples were treated with RNase A, purified DNA fragments were separated by gel electrophoresis, and mononucleosome bands were gel purified. MNase-treated and input samples were quantified by qPCR on a 7500 FAST Real-Time PCR machine (Life Technologies) using SYBR green mix (Life Technologies). The scanning primer pairs covering the *IRT2* locus and upstream region used for the analysis are available in Supplementary Table 2.

**Northern blotting**. We adapted a northern blot protocol that was described previously[57]. In short, total RNA was extracted with acid phenol:chloroform:isoamyl alcohol (125:24:1) and precipitated in ethanol with 0.3 M sodium acetate. RNA samples were denatured in a glyoxal/DMSO mix (1 M deionized glyoxal, 50% v/v DMSO, 10 mM NaPi buffer pH 6.5–6.8) at 70 °C for 10 min. Denatured samples were mixed with loading buffer (10% v/v glycerol, 2 mM NaPi buffer pH 6.5–6.8, 0.4% w/v bromophenol blue) and separated on an agarose gel (1.1% w/v agarose, 0.01 M NaPi buffer) for at least 2 h at 80 V. RNAs were then transferred onto nylon membranes overnight by capillary transfer. The ribosomal RNA bands were visualized by methylene blue staining. The membranes were blocked for at least 3 h

at 42 °C in hybridization buffer (1% w/v SDS, 50% v/v deionized formamide, 25% w/v dextran sulfate, 58 g/l NaCl, 200 mg/l herring sperm single strand DNA, 2 g/l BSA, 2 g/l polyvinyl-pyrolidone, 2 g/l ficoll, 1.7 g/l pyrophosphate, 50 mM Tris pH 7.5) before hybridization. The radioactive probes were synthesized using a Prime-It II Random Primer Labeling Kit (Agilent), a target-specific DNA template and dATP, [α-32P] (Perkin-Elmer). The oligo nucleotide sequences used to generate target-specific DNA template for amplifying the northern blot probes are displayed in Supplementary Table 2. A representative blot of at least two experimental repeats was used in all figures. The full size uncropped blots are displayed in Supplementary Figure 12.

**Single-molecule RNA fluorescent in situ hybridization**. The smRNAFISH was performed as described previously[2,58]. In short, cells were fixed with 3% (w/v) formaldehyde overnight, treated with zymolyase and further fixed in 80% ethanol. Subsequently cells were hybridized for at least 16 h at 30 °C in a buffer (10% v/v formamide, 10% w/v dextran sulfate, 2 mM ribonucleoside vanadyl complexes, 0.02% w/v RNAse-free BSA, 1 mg/ml *Escherichia coli* transfer RNAs, 0.3 M sodium chloride, 0.03 M sodium citrate) containing fluorophore labeled probes directed to *IME1* (AF594) and the internal control *ACT1* (Cy5). Cells were pelleted after addition of 1 ml of wash buffer (10% formamide, 0.3 M sodium chloride, 0.03 M sodium citrate), re-suspended in 400 μl of Dapi buffer (5 μg/ml 4′,6-diamidino-2-phenylindole, 10% formamide, 0.3 M sodium chloride, 0.03 M sodium citrate) and incubated 30 min at 30 °C. Cells were then pelleted, re-suspended in 1 ml of wash buffer, incubated 30 min at 30 °C before being pelleted, and re-suspended in 200 μl of 0.3 M sodium chloride, 0.03 M sodium citrate. Cells were imaged using a ×100 oil objective, Numerical aperture (NA) 1.4, on a Nikon TI-E imaging system (Nikon). DIC, DAPI, AF594 (*IME1*), Cy5 (*ACT1*) images were collected every 0.3 micron (20 stacks) using an ORCA-FLASH 4.0 camera (Hamamatsu) and NIS-element software (Nikon). ImageJ software was used to make maximum intensity Z projections of the images[59]. Subsequently, StarSearch software (http://rajlab.seas.upenn.edu/StarSearch/launch.html, Raj laboratory, University of Pennsylvania) was used to quantify transcripts in single cells. Comparable thresholds were used to count RNA foci in single cells. Only cells positive for the internal control *ACT1* were used for the analysis. At least a total $n = 120$ cells were counted for each experiment.

**β-Galactosidase assay**. Liquid ortho-nitrophenyl-β-galactoside (ONPG) assay was performed as described previously[2]. In short, 1 ml of $OD_{600} = 1.8$ cells were pelleted and washed once with buffer Z (Phosphate buffer pH 7, KCl 10 mM, MgCl 1 mM) and cells were snap-frozen in liquid nitrogen. Cells were chemically disrupted using Y-PER buffer (Thermo Scientific), cells were incubated with ONPG (Sigma) (1 mg/ml in Z buffer plus 50 mM β-mercaptoethanol) till yellow coloring occurred. The reaction was quenched using sodium carbonate (1 mM) and cell debris were cleared by centrifugation. Absorption of each sample was measured at $OD_{420}$ using a 96 wells plate reader. Miller units were calculated according to a standard formula: Miller unit = (signal from plate reader ($OD_{420}$) × 1000)/(cell density ($OD_{600}$) × time of incubation with ONPG (min)). The data from the experiments represents the SEM of at least two biological experiments.

**Fluorescence GFP quantification**. For Fig. 4e and Supplementary Figure 6, cells were fixed with 3% (w/v) formaldehyde for 10 min at room temperature, cells were washed once with phosphate-sorbitol buffer (0.1 M kPi (pH 7), 1.2 M sorbitol), re-suspended in phosphate-sorbitol buffer, and stored till imaging at 4 °C. Imaging was performed using a ×100 oil objective, NA 1.4, on a Nikon TI-E imaging system (Nikon), and 500 ms exposure time for the GFP channel. Images were collected using an ORCA-FLASH 4.0 camera (Hamamatsu) and NIS-element software (Nikon). ImageJ software was used for the quantification[59]. The fluorescence signals were determined by quantifying the whole-cell fluorescence, which was corrected for the background fluorescence signal and auto-fluorescence signal from wild-type cells. The signals represented in Fig. 4e and Supplementary Figure 6 are relative to the 1 h time point.

**Live-cell imaging**. Cells were imaged using a Zeiss Observer Z1 microscope equipped with a motorized stage, automatic focusing hardware, and temperature control with a ×40 (Zeiss EC Plan-Neofluar ×40/1.3 Ph3 WD = 0.21 M27, oil) phase objective every 12 min. Exposure times were 250 ms for GFP and 120 ms for mCherry using the Zeiss Kolibri LED light system with the 470 nm and 540–580 LEDs at 25% intensity was used with the following filter sets for GFP: excitation: FF01-504/12-25, dichroic: FF518-Di01-25x36, emission: FF01-530/11-25 and for mCherry: excitation: FF01-563/9-25, dichroic: FF573-Di01-25x36, emission: FF01-598/25-25, (all filters from Semrock). Cells were segmented and tracked using a previously published algorithm[60]. Fluorescent concentrations were calculated by subtracting background signals from the segmented cells and dividing with the approximated cell volumes (cell area). Activation times and slopes were calculated using custom MATLAB software (available upon request). GFP-Ime1 onset time was determined relative to *pIME1-Δirt2*-mCherry onset that was used to define the time 0. The activation rates were calculated by fitting a line to the concentration curve after the onset and finding its slope.

**Determining the proximity of CUTs and SUTs to coding genes**. For Supplementary Figure 11, a data set from Wery et al.[61] was used. For the analysis. we searched for genes that showed expression of *CUTs* or *SUTs* within 400 bp upstream or downstream of the coding sequence (±400 bp). An overlap of 1 bp or more between a gene ±400 bp and a *CUTs* or *SUTs* was considered a positive hit in our analysis.

**Data availability**. The data generated in the current study are available from the corresponding author.

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

## Acknowledgements

We are grateful to Michael Knop for the GFP tagging plasmid, and Frank Uhlmann, Paola Scaffidi, Jessica Greenwood, and the members of the CGR lab for their critical reading of the manuscript. This work was supported by CPRIT (RR150058) & the Welch foundation (I-1919-20170325) to A.D. and by the Francis Crick Institute to F.M. and F.J. v.W., which receives its core funding from Cancer Research UK (FC001203), the UK Medical Research Council (FC001203), and the Wellcome Trust (FC001203).

## Author contributions

F.M. and F.J.v.W. conceived and designed the study. F.M. performed the experiments. N. E.W. and A.D. performed the time-lapse microscopy experiments, analyzed data, and provided the mathematical model. G.K. performed bioinformatics analyses for Supplementary Figures 1 and 11. F.M and F.J.v.W. analyzed data. F.M. and F.J.v.W. wrote the paper.

## Additional information

**Competing interests:** The authors declare no competing financial interests.

