## [Peer Review File · Nature Communications]

Reviewers' comments:

Reviewer #1 (Remarks to the Author):

Manuscript ID: NCOMMS-17-23482-T

Title: A regulatory circuit of two contiguous lncRNAs and a master regulator directs cell fate in yeast.

Transcription of lncRNAs is known to contribute to gene regulation at specific genomic loci by changing local chromatin structure. Previously, it was shown that IRT1 transcription establishes a repressive chromatin over IME1 promoter regions by recruiting Set2 HMT and Set3 HDAC. This study aims to understand the role of IRT2 transcription expressing an additional lncRNA from upstream regions of IRT1 on IME1 regulation. This study shows that transcription of these two contiguous lncRNAs fine-tunes timing and rate of IME1 accumulation. Furthermore, Ime1 itself activates its own transcription by activating IRT2 transcription that down-regulates IRT1 transcription. Overall, this study is nicely done and the results will be of interest to a wide audience. Prior to publication, the following should be addressed.

1. In Figure 3, Ume6 directly binds to upstream regions of IRT2 and activates IRT2 transcription. The authors stated that although Ume6 normally acts as a repressor, it can function as an activator in the presence of Ime1. In discussion section, the authors also mentioned that the expression of IRT2 and genes required for meiosis depend on a physical interaction between Ime1 and Ume6. I might have missed it but where is this shown in the cited paper? The authors should test whether Ime1 directly binds to IRT2 promoter and whether this binding requires Ume6. Furthermore, it should be tested whether Ume6 binding to IRT2 promoter is affected by loss of Ime1.
2. To confirm whether Ime1 activates IRT2 transcription, the authors measured IRT2 levels in WT and ime1-t mutant (Figure 3c). It should be tested whether ime1-t mutation does not affect IRT2 transcription in pCUP-IRT2(RME1-H) strain background (Figure 2a) because IRT2 transcription from pCUP could be independent of Ume6 and Ime1.
3. A previous study from the corresponding author revealed that IRT1 transcription recruits Set2 HMT and Set3 HDAC to repress IME1 expression in haploid cells. Since IRT1 is also active in diploid cells, it would be interesting to test whether loss of Set2 and Set3 increases IME1 and IRT2 transcription.
4. Overall, the texts and figure legends require more detailed explanations.
5. Figure 1d and 1e are not very informative. Schematic overview of the strains used including pCUP-IRT2 and RME1-H should be included.
6. It would be better to include the Rme1 activator in Figure 7.

Reviewer #2 (Remarks to the Author):

In this manuscript, the authors investigate the regulatory mechanism behind IME1 expression in yeast meiosis. They demonstrate the existence of a feedback loop involving two contiguous non-coding RNAs (ncRNAs). Expression of IME1 activates IRT2, which then interferes with the expression of IRT1, thereby relieving IRT1-mediated inhibition of IME1. IRT1 and IME1 expression control is mediated through a changed chromatin environment generated by transcription. Non-coding regulation of gene expression is well-explored, as are transcription-coupled chromatin changes. The primary novelty of the system investigated is that it employs more than one ncRNA. Even here, however, systems with more than one ncRNA are known (e.g. FLO11, as pointed out by the authors in the Discussion). However, the arrangement here does seem to be novel. Nevertheless I am not completely persuaded that the set-up here is sufficiently novel to permit publication in Nature Comms. Moreover, I think the authors have missed a trick in not applying more quantitative methodologies to this system. A considerable amount of quantitative data is

amassed, but there is no attempt to develop a quantitative model that could potentially explain this data. Such a model could potentially highlight gaps in the conceptual understanding of the system, always a real possibility in complex systems involving feedback.

I have another comment concerning the nature of the changed chromatin environment induced by transcription: at present the authors only assay nucleosome positioning via MNase and qPCR. It would probably be helpful to also assay other chromatin features, e.g. histone modifications, that are indicative of active or repressed transcriptional activity.

Finally, I think some effort to understand why this system uses such a complex set of ncRNAs to mediate transcriptional upregulation of IME1 would be useful. Why doesn't the cell employ more direct positive feedback, would that be too fast?

Reviewers' comments:

Reviewer #1 (Remarks to the Author):

Manuscript ID: NCOMMS-17-23482-T

Title: A regulatory circuit of two contiguous lncRNAs and a master regulator directs cell fate in yeast.

Transcription of lncRNAs is known to contribute to gene regulation at specific genomic loci by changing local chromatin structure. Previously, it was shown that IRT1 transcription establishes a repressive chromatin over IME1 promoter regions by recruiting Set2 HMT and Set3 HDAC. This study aims to understand the role of IRT2 transcription expressing an additional lncRNA from upstream regions of IRT1 on IME1 regulation. This study shows that transcription of these two contiguous lncRNAs fine-tunes timing and rate of IME1 accumulation. Furthermore, Ime1 itself activates its own transcription by activating IRT2 transcription that down-regulates IRT1 transcription. Overall, this study is nicely done and the results will be of interest to a wide audience. Prior to publication, the following should be addressed.

1. In Figure 3, Ume6 directly binds to upstream regions of IRT2 and activates IRT2 transcription. The authors stated that although Ume6 normally acts as a repressor, it can function as an activator in the presence of Ime1. In discussion section, the authors also mentioned that the expression of IRT2 and genes required for meiosis depend on a physical interaction between Ime1 and Ume6. I might have missed it but where is this shown in the cited paper?

The authors should test whether Ime1 directly binds to IRT2 promoter and whether this binding requires Ume6.

We thank the reviewer for the comment and suggestion. The statement “*IRT2* and genes required for meiosis depend on a physical interaction between Ime1 and Ume6” is based on published work from several other groups. It has been well established that Ume6 and Ime1 can interact with each other¹⁻³. This interaction relies on the activity of nutrient sensing kinases¹⁻³. However, there is no evidence that Ime1 directly binds to Ume6 on chromatin. We have attempted to measure Ime1 binding to Ume6 site upstream in the *IME1* promoter by chromatin immunoprecipitation (ChIP). We have tried several affinity tags fused to *IME1* (GFP/HA/V5) and multiple crosslinking conditions. Unfortunately, we were not able to detect Ime1 binding at the Ume6 site. We speculate that the Ime1 ChIP assay is not sensitive enough because of the following reasons: (1) Ime1 itself does not directly bind DNA, hence it will be difficult to generate formaldehyde crosslink sites between Ime1 and DNA, (2) the Ime1 protein has a high turnover rate (a few minutes) suggesting that the Ime1-Ume6 interaction is transient⁴. To capture the Ume6-Ime1 interaction by ChIP at the Ime1 promoter protein-protein cross-linkers may need to be tested or other specialized assays need to be developed, which we think is beyond the scope of the current manuscript.

Given that we were not able to show the interaction in current manuscript, we have changed the text in the discussion to “It has been shown that Ime1 and Ume6 can physically interact with each other^{1,3}. The Ume6-Ime1 interaction is required for the activation of early meiotic genes³.”

Furthermore, it should be tested whether Ume6 binding to IRT2 promoter is affected by loss of Ime1.

We have performed this control experiment and this is now included in Supplementary Figure 4a. We show that Ume6 binding is not affected by loss of Ime1 prior and upon starvation.

2. To confirm whether Ime1 activates IRT2 transcription, the authors measured IRT2 levels in WT and *ime1-t* mutant (Figure 3c). It should be tested whether *ime1-t* mutation does not affect IRT2 transcription in pCUP-IRT2(RME1-H) strain background (Figure 2a) because IRT2 transcription from pCUP could be (in)dependent of Ume6 and Ime1.

We have included this control experiment in Supplementary Figure 3e. We show that *IRT2* transcription is not affected by *ime1-t*. In addition, we show that *IRT2* mediated repression of *IRT1* is not affected by *ime1-t*.

3. A previous study from the corresponding author revealed that IRT1 transcription recruits Set2 HMT and Set3 HDAC to repress IME1 expression in haploid cells. Since IRT1 is also active in diploid cells, it would be interesting to test whether loss of Set2 and Set3 increases IME1 and IRT2 transcription.

We thank the reviewer for the suggestion. We have included this experiment in Figure 4d. We generated the *set2/set3* double mutant in the *RME1-H* background in diploid cells, performed a meiotic time course and measured *IME1*, *IRT1*, and *IRT2* expression. Since *IRT1*-mediated repression of *IME1* is not functional, *IME1* expression is increased in the *set2/set3* double mutant⁵. As expected, *IRT2* expression is also increased because Ime1 is increased. Interestingly, we find that *IRT1* expression is repressed and its expression anti-correlates with *IRT2* expression. This suggests that Set2 and Set3 are not important for *IRT2* mediated repression of *IRT1*. In line with this observation, we find that Set2 dependent histone 3 lysine 36 methylation is not present in the *IRT2* region of the *IME1* promoter even the presence of *IRT2* transcription (Supplementary Figure 5b). Finally, we show that *IRT2* mediated repression of *IRT1* is functional in the *set2/set3* mutant when expressed *IRT2* is expressed from the *CUP1* promoter (*pCUP-IRT2*) (Supplementary Figure 5c).

4. Overall, the texts and figure legends require more detailed explanations.

We went over the legends and text and provided more explanation where needed. Please note that we are restricted in the number of words used for the Figure legends because of the journal's format.

5. Figure 1d and 1e are not very informative. Schematic overview of the strains used including pCUP-IRT2 and RME1-H should be included.

We have included a scheme of the genotypes used for Figure 1e and 1f (old Figure 1d and 1e) in separate panel Figure 1d. We think this makes the experiment more informative.

6. It would be better to include the Rme1 activator in Figure 7.

We have now included Rme1 in the model of Figure 7e (old figure 7).

Reviewer #2 (Remarks to the Author):

In this manuscript, the authors investigate the regulatory mechanism behind IME1 expression in yeast meiosis. They demonstrate the existence of a feedback loop involving two contiguous non-coding RNAs (ncRNAs). Expression of IME1 activates IRT2, which then interferes with the expression of IRT1, thereby relieving IRT1-mediated inhibition of IME1. IRT1 and IME1 expression control is mediated through a changed chromatin environment generated by transcription. Non-coding regulation of gene expression is well-explored, as are transcription-coupled chromatin changes. The primary novelty of the system investigated is that it employs more than one ncRNA. Even here, however, systems with more than one ncRNA are known (e.g. FLO11, as pointed out by the authors in the Discussion). However, the arrangement here does seem to be novel. Nevertheless I am not completely persuaded that the set-up here is sufficiently novel to permit publication in Nature Comms. Moreover, I think the authors have missed a trick in not applying more quantitative methodologies to this system. A considerable amount of quantitative data is amassed, but there is no attempt to develop a quantitative model that could potentially explain this data. Such a model could potentially highlight gaps in the conceptual understanding of the system, always a real possibility in complex systems involving feedback.

We thank the reviewer for the suggestion. We have now included a simple mathematical model in which we examined the dynamic properties of the regulatory circuit. We added a separate section in the main and supplemental text that describes the model (Fig. 7, Supplementary Fig. 10 and Supplementary information).

In the model we describe the transcription rates of *IRT1*, *IRT2*, and *IME1* as well as *IME1* mRNA and Ime1 protein levels (Fig. 7b and Supplementary Fig. 10a). In our model the starvation signal, which is the primary signal for activation of *IRT1* and *IME1* transcription, was considered to be either off (0) or on (1) (Fig. 7b)^{6,7}. We think that this simple model is sufficient to reflect the experimental data. For example, in the absence of *IRT1* and *IRT2* transcription, Ime1 levels accumulated faster and were higher compared to the wild type, in line with the observations from the single cell analyses (compare Fig. 5c (middle and right panel) to Fig. 7b (middle and right panel)). Conversely, in the absence of *IRT2* transcription Ime1 levels were reduced compared to wild type. The model predicts a gradual increase in *IME1* transcription rates during starvation, which is in line with our observation that Ime1 can stimulate its own expression via *IRT2* and *IRT1* (Fig. 7b, middle panel).

We also used our model to simulate how *IRT2* and *IRT1* control the dynamics of Ime1 accumulation during different periods of starvation (Fig. 7c and Supplementary Fig. 10b). We find that prolonged starvation (5 hours) lead to almost maximum Ime1 expression for the wild-type *IME1* promoter, while a short period of starvation (1 hour) resulted in low levels of Ime1 accumulation (Fig. 7c, compare left and right panels to Fig. 7b right panel). In the absence of *IRT1* and *IRT2*, a short period of starvation (1 hour) displayed comparable Ime1 levels to a simulation of the wild type starved for 5 hours (Fig. 7c, compare yellow left panel to blue line right panel). This suggests that the *IRT2-IRT1-Ime1* circuit restricts Ime1 expression in cells subjected to starvation for short time periods. Taken together, we propose that there are possibly two functions for the feedback loop: (1) to elevate the rate of *IME1* transcription during prolonged starvation, (2) to prevent mis-expression of Ime1 when the cells starved for a short of time.

I have another comment concerning the nature of the changed chromatin environment induced by transcription: at present the authors only assay nucleosome positioning via MNase and qPCR. It would probably be helpful to also assay other chromatin features, e.g. histone modifications, that are indicative of active or repressed transcriptional activity.

We thank the reviewer for the comment and suggestion. In budding yeast several pathway have been implicated in noncoding transcription coupled repression of gene expression⁸. In particular, the Set2/Rpd3S and Set1/Set3C pathways have shown to be important for gene repression by noncoding transcription^{5,9,10}. In short, noncoding transcription deposits Set2-dependent histone 3 lysine 36 methylation and Set1 dependent histone H3 lysine 4 dimethylation. These marks are recognized by Rpd3S and Set3C histone deacetylases^{11,12}. For example, this mechanism is used for *IRT1*-mediated repression of *IME1*, but also at many other loci^{5,9,10}.

To examine whether *IRT2*-mediated repression of *IRT1* also requires Set2/Rpd3S and Set1/Set3C pathway, we generated the *set2/set3* double mutant in the *RME1-H* background in diploid cells, and performed a meiotic time course. Since *IRT1*-mediated repression of *IME1* is not functional, *IME1*

expression is increased in the *set2/set3* double mutant⁵. As expected, *IRT2* expression is also increased because *Ime1* is increased. Interestingly, we find that *IRT1* expression is repressed and its expression anti-correlates with *IRT2* expression. This suggests that *Set2* and *Set3* are not important for *IRT2* mediated repression of *IRT1*. In line with this observation, we find that *Set2* dependent histone 3 lysine 36 methylation is not present in the *IRT2* region of the *IME1* promoter even when *IRT2* is transcribed (Supplementary Figure 5b). Finally, we show that *IRT2* mediated repression of *IRT1* is functional in the *set2/set3* mutant when expressed from *pCUP-IRT2*.

Finally, I think some effort to understand why this system uses such a complex set of ncRNAs to mediate transcriptional upregulation of *IME1* would be useful. Why doesn't the cell employ more direct positive feedback, would that be too fast?

We agree that the question why yeast cells have evolved to a system comprising of two noncoding transcripts for the regulation of *IME1* is interesting. We think the question has to be dissected in two parts. Why *IRT1* and not a direct repressor is needed for *IME1* repression? Why does *Ime1* feeds back to its promoter via *IRT2* and not directly?

Why *IRT1* and not a direct repressor is needed for *IME1* repression? The answer to this question was described previously⁵. In short, we proposed that the main reason for *IRT1* mediated repression of the *IME1* promoter is that the *IME1* promoter is relative large over 1.5 kb, which will require multiple sequence specific repressors for efficient repression of the whole *IME1* promoter. Instead transcription of a single transcript can repress the *IME1* promoter, which only requires the *Rme1* binding sites upstream in the *IME1* promoter. Transcription of the non-coding RNA serves/allows transmission of the repressive signal over a relative large promoter region.

The next question is why *IRT2* and not *Ime1* directly facilitates the feedback regulation? In case *IME1* feeds back to its own promoter, there is a possible constraint. Given that *IRT1* is transcribed, a feedback of *Ime1* to its own promoter will somehow need to bypass *IRT1* mediated repression of *IME1*, which is not possible if transcription of *IRT1* is not repressed. We propose that repression of *IRT1* is needed to obtain maximum levels of *IME1* activation.

We have added the following section to the discussion of the manuscript:

“Why does *Ime1* feed back to its own promoter via *IRT2*? Previous work showed that transcription of *IRT1* represses a relatively large region (over 1.5 kb) of the *IME1* promoter requiring the upstream *Rme1* binding sites⁵. If *Ime1* feeds back to its own promoter downstream of the *Rme1* binding sites

to directly activate its transcription, it would face interference by transcription of *IRT1*. Our data and modelling shows that repression of *IRT1* transcription facilitates full activation of the *IME1* promoter. We propose that transcription of *IRT2* is an effective way of repressing *IRT1* mediated interference of the *IME1* promoter, and thereby promotes Ime1 accumulation and entry into meiosis.

References:

1. Malathi, K., Xiao, Y. & Mitchell, A.P. Interaction of yeast repressor-activator protein Ume6p with glycogen synthase kinase 3 homolog Rim11p. *Mol Cell Biol* **17**, 7230-6 (1997).
2. Nakazawa, N., Abe, K., Koshika, Y. & Iwano, K. Cln3 blocks IME1 transcription and the Ime1-Ume6 interaction to cause the sporulation incompetence in a sake yeast, Kyokai no. 7. *J Biosci Bioeng* **110**, 1-7 (2010).
3. Rubin-Bejerano, I., Mandel, S., Robzyk, K. & Kassir, Y. Induction of meiosis in *Saccharomyces cerevisiae* depends on conversion of the transcriptional repressor Ume6 to a positive regulator by its regulated association with the transcriptional activator Ime1. *Mol Cell Biol* **16**, 2518-26 (1996).
4. Chia, M. *et al.* Transcription of a 5' extended mRNA isoform directs dynamic chromatin changes and interference of a downstream promoter. *Elife* **6**(2017).
5. van Werven, F.J. *et al.* Transcription of two long noncoding RNAs mediates mating-type control of gametogenesis in budding yeast. *Cell* **150**, 1170-81 (2012).
6. Moretto, F. & van Werven, F.J. Transcription of the mating-type-regulated lncRNA *IRT1* is governed by TORC1 and PKA. *Curr Genet* **63**, 325-329 (2017).
7. van Werven, F.J. & Amon, A. Regulation of entry into gametogenesis. *Philos Trans R Soc Lond B Biol Sci* **366**, 3521-31 (2011).
8. Venkatesh, S. & Workman, J.L. Histone exchange, chromatin structure and the regulation of transcription. *Nat Rev Mol Cell Biol* **16**, 178-89 (2015).
9. Kim, T., Xu, Z., Clauder-Munster, S., Steinmetz, L.M. & Buratowski, S. Set3 HDAC mediates effects of overlapping noncoding transcription on gene induction kinetics. *Cell* **150**, 1158-69 (2012).
10. Kim, J.H. *et al.* Modulation of mRNA and lncRNA expression dynamics by the Set2-Rpd3S pathway. *Nat Commun* **7**, 13534 (2016).
11. Keogh, M.C. *et al.* Cotranscriptional Set2 methylation of histone H3 lysine 36 recruits a repressive Rpd3 complex. *Cell* **123**, 593-605 (2005).
12. Kim, T. & Buratowski, S. Dimethylation of H3K4 by Set1 recruits the Set3 histone deacetylase complex to 5' transcribed regions. *Cell* **137**, 259-72 (2009).

REVIEWERS' COMMENTS:

Reviewer #1 (Remarks to the Author):

The authors addressed nearly all the points raised by reviewers in a satisfying way. I believe this manuscript is now suitable for publication in Nature Communications.

Reviewer #2 (Remarks to the Author):

The authors have revised the paper to incorporate many of the changes I requested during the first round. I think these additions have improved the paper, with the additional model and discussion helping to elucidate the underlying mechanisms. Other than a small typo in Eq. 5 in the SI, I think the revisions are fine and I would be happy to see the manuscript published.

Please find below our response to reviewers' comments.

REVIEWERS' COMMENTS:

Reviewer #1 (Remarks to the Author):

The authors addressed nearly all the points raised by reviewers in a satisfying way. I believe this manuscript is now suitable for publication in Nature Communications.

Great.

Reviewer #2 (Remarks to the Author):

The authors have revised the paper to incorporate many of the changes I requested during the first round. I think these additions have improved the paper, with the additional model and discussion helping to elucidate the underlying mechanisms. Other than a small typo in Eq. 5 in the SI, I think the revisions are fine and I would be happy to see the manuscript published.

We have corrected the typo in equation 5. It now says l/m instead of l .